# Reading Your Heart: Learning ECG Words and Sentences via Pre-training ECG Language Model

**Jiarui Jin**[1,2,3,4,*], **Haoyu Wang**[1,4,*], **Hongyan Li**[2,3,†], **Jun Li**[1], **Jiahui Pan**[4,†], **Shenda Hong**[1,†]
[1]National Institute of Health Data Science, Peking University
[2]State Key Laboratory of General Artificial Intelligence, Peking University
[3]School of Intelligence Science and Technology, Peking University
[4]School of Artificial Intelligence, South China Normal University
hongshenda@pku.edu.cn, panjiahui@m.scnu.edu.cn, leehy@pku.edu.cn

## Abstract

Electrocardiogram (ECG) is essential for the clinical diagnosis of arrhythmias and other heart diseases, but deep learning methods based on ECG often face limitations due to the need for high-quality annotations. Although previous ECG self-supervised learning (eSSL) methods have made significant progress in representation learning from unannotated ECG data, they typically treat ECG signals as ordinary time-series data, segmenting the signals using fixed-size and fixed-step time windows, which often ignore the form and rhythm characteristics and latent semantic relationships in ECG signals. In this work, we introduce a novel perspective on ECG signals, treating heartbeats as words and rhythms as sentences. Based on this perspective, we first designed the QRS-Tokenizer, which generates semantically meaningful ECG sentences from the raw ECG signals. Building on these, we then propose HeartLang, a novel self-supervised learning framework for ECG language processing, learning general representations at form and rhythm levels. Additionally, we construct the largest heartbeat-based ECG vocabulary to date, which will further advance the development of ECG language processing. We evaluated HeartLang across six public ECG datasets, where it demonstrated robust competitiveness against other eSSL methods. Our data and code are publicly available at https://github.com/PKUDigitalHealth/HeartLang.

## 1 Introduction

Electrocardiogram (ECG) is a common type of clinical data used to monitor cardiac activity, and is frequently employed in diagnosing cardiac diseases or conditions impairing myocardial function (Hong et al., 2020; Liu et al., 2021). A primary limitation of using supervised deep learning methods for ECG signal analysis is their dependency on large-scale, expert-reviewed, annotated high-quality data. Moreover, even with sufficient data, these methods are often designed to address specific tasks, which curtails the generalization ability

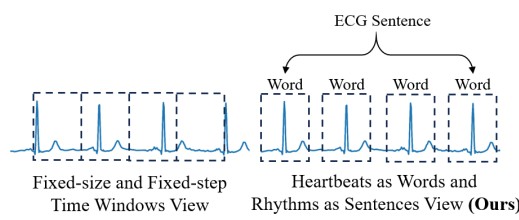

Figure 1: Two perspectives on ECG signals.

of the model. To overcome these challenges, ECG self-supervised learning (eSSL) has demonstrated efficacy by training on vast amounts of unlabeled ECG recordings to learn generic ECG signal representations, which are then fine-tuned for specific downstream tasks (Pup & Atzori, 2023).

---

*Equal contribution.
†Corresponding authors.

Current eSSL methods can be primarily classified into two categories: contrastive-based methods and reconstruction-based methods. The core principle of contrastive-based methods involves creating positive and negative sample pairs, aiming to maximize the similarity of positive pairs and minimize the similarity of negative pairs (Zhang et al., 2023b). Reconstruction-based methods focus on training a model to reconstruct the original input from partial or transformed data, thus learning effective data representations (Zhang et al., 2023c). However, almost all methods treat ECG signals as ordinary time-series data, which have two significant drawbacks:

**Ignoring Form and Rhythm Characteristics of ECG.** ECG diagnostics from multi-level characteristics are essential Hong et al. (2019). For example, myocardial infarction is diagnosed by observing ST segment elevation of a single heartbeat (Vogel et al., 2019). Likewise, cardiac rhythm characteristics are critical, as arrhythmias like atrial fibrillation (AF) are identified based on the overall cardiac rhythm (Carrington et al., 2022). However, existing eSSL methods typically employ fixed-size and fixed-step time windows to segment the signal (Song et al., 2024). This perspective treats ECG signals as ordinary time-series signals, thereby ignoring the unique form and rhythm characteristics inherent to ECG signals, ultimately leading to a decline in the effectiveness of self-supervised learning for both.

**Ignoring Latent Semantic Relationships of ECG.** Due to significant differences in heart rate and other factors between different subjects, and even among different samples from the same subject (Lan et al., 2022), using fixed-size and fixed-step time windows to segment data leads to substantial discrepancies among samples. The differences between samples disrupt the potential semantic relationships between different heartbeats, which in turn negatively impacts the effectiveness of learning a generalized representation in self-supervised learning.

To address these challenges, we propose a self-supervised learning framework named **HeartLang** for ECG language processing (ELP). A distinguishing feature of ECG signals is the clear visibility of heart rate patterns, where individual heartbeats easily identifiable. The core concept of our framework treats heartbeats as words and rhythms as sentences, enabling self-supervised learning at both form and rhythm levels to capture multi-level general representations. Our method consists of four key components: (1) the QRS-Tokenizer, which generates the ECG sentences from the raw ECG signals; (2) the ST-ECGFormer, which leverages spatio-temporal information to enhance latent semantic extraction from ECG sentences; (3) the construction of the largest ECG vocabulary to date, where heartbeat quantization and reconstruction enable form-level representation learning; and (4) masked ECG sentence pre-training, which facilitates rhythm-level general representation learning. Through these approaches, our method can learn both form-level and rhythm-level representations of ECG signals without labels, and extract latent semantic representations in ECG sentences. The main contributions of this work are summarized in below:

- We propose HeartLang, a novel self-supervised learning framework for ECG language processing, designed to learn general representations at form and rhythm levels and extract latent semantic relationships from unlabeled ECG signals.
- We present a paradigm-shifting perspective of ECG signals, treating them as a language with distinct words (heartbeats) and sentences (rhythms), and design a QRS-Tokenizer that generates the ECG sentences from the raw ECG signals based on this perspective.
- We design ST-ECGFormer, a novel transformer-based backbone network for ECG signal analysis, which leverages the spatio-temporal features in ECG signals to enhance representation learning and optimize latent semantic relationships extraction for ECG sentences.
- To the best of our knowledge, we have constructed the largest ECG vocabulary based on heartbeats to date. This ECG vocabulary includes a wide variety of heartbeat morphological representations across different cardiac conditions, which will further advance the development of ECG language processing.

## 2 RELATED WORK

### 2.1 SELF-SUPERVISED LEARNING FOR ECG SIGNALS

In recent years, ECG self-supervised learning (eSSL) has demonstrated its ability to learn general representations from unlabeled ECG signals, significantly improving the performance of down-

stream tasks (Lai et al., 2023). eSSL methods can be broadly categorized into two types: contrastive-based methods and reconstruction-based methods. For contrastive-based approaches, CLOCS (Kiyasseh et al., 2021) enhances contrastive learning by leveraging cross-space, time, and patient-level relationships in ECG signals, while ASTCL (Wang et al., 2024) employs adversarial learning to capture spatio-temporal invariances in ECG signals. ISL (Lan et al., 2022) enhances cross-subject generalization ability through inter-subject and intra-subject contrastive learning, while BTFS (Yang & Hong, 2022) enhances ECG signal classification performance by combining time-domain and frequency-domain contrastive learning. On the other hand, reconstruction-based methods like MaeFE (Zhang et al., 2023a) and ST-MEM (Na et al., 2024) adopt a spatio-temporal approach, learning general ECG representations by masking and reconstructing temporal or spatial content. CRT (Zhang et al., 2023c) obtains general representations in ECG signals by mutually reconstructing the time-domain and frequency-domain data. However, existing eSSL methods predominantly focus on spatio-temporal or time-frequency domain representation learning of ECG signals, treating them as ordinary time-series data. This perspective often neglects the morphologically rich semantic information embedded in individual heartbeats.

## 2.2 ECG LANGUAGE PROCESSING

ECG language processing (ELP) is an emerging paradigm for handling ECG signals, first proposed by Mousavi et al. (2021). Since ECG signals inherently possess significant and clear semantic information in heartbeats, they can be processed using methods similar to natural language processing (NLP). Both Mousavi et al. (2021) and Choi et al. (2023) employ approaches that segment different waves within heartbeats to construct vocabularies for modeling. However, when dealing with ECG signals of varying quality, existing methods struggle to accurately segment fine-grained waveforms. Moreover, current ELP methods have relatively small vocabularies (no more than 70 clusters), which limits the richness of the semantic information. In addition, research on ELP remains sparse, highlighting it as a field in urgent need of further exploration. To address these limitations, we propose a new perspective that directly treats heartbeats as words for modeling and have built the largest ECG vocabulary to date, consisting of 5,394 words, which will significantly advance the development of the ELP research field.

## 3 METHOD

In this section, we provide a detailed explanation of the specific structure of the HeartLang framework. We first define multi-lead ECG data as $X \in \mathbb{R}^{C \times T}$, where $C$ represents the number of ECG leads (electrodes) and $T$ represents the total timestamps. The configuration of ECG leads follows the standard 12-lead ECG setup. The overview of the framework is shown in Figure 2. The use of the framework can be divided into four steps. First, a QRS-Tokenizer is used to generate the ECG sentences from the raw ECG signals as described in the Section 3.1. Second, constructing the ECG vocabulary is achieved through the steps in the Section 3.3. Third, masked ECG sentence pre-training of the framework is performed as described in the Section 3.4. Finally, fine-tuning is performed for downstream tasks.

### 3.1 GENERATING ECG SENTENCES USING THE QRS-TOKENIZER

**QRS Detection.** A key concept of our method is to treat heartbeats as words, thus making the segmentation of the original ECG signal into semantic heartbeat patches essential. We introduce QRS-Tokenizer, a tokenizer that generates the ECG sentences from the raw ECG signals based on QRS waveforms. Initially, the I-lead signal is bandpass filtered between 5 and 20 Hz, followed by moving wave integration (MWI) using a Ricker wavelet on the filtered signal, and the squared integration signal is saved. The local maxima of the MWI signal are then traversed, with each maximum that occurs after the refractory period and exceeds the QRS detection threshold being classified as a QRS complex. Following detection, we obtain the indices of the detected QRS complexes $Q = \{q_i | i = 1, \ldots, N\}$, where $N$ denotes the number of detected QRS indices per sample, which varies between ECG recordings.

Assuming the time window size is $t$. For each lead, we center each index in $Q$, using the midpoint between every two indices as the interval boundaries, and independently segment the QRS complex

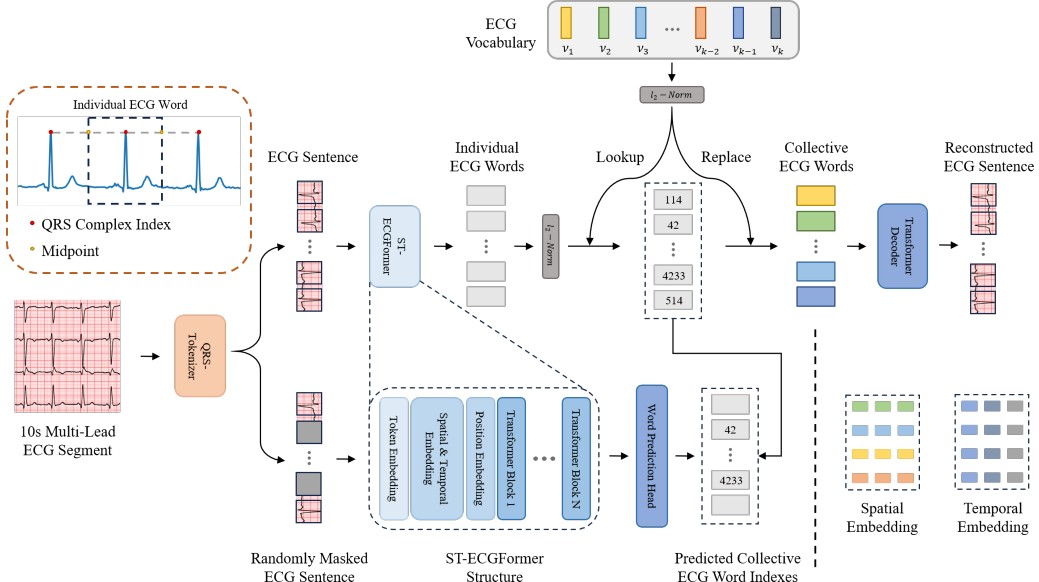

Figure 2: Framework of HeartLang.

patches for each lead. If the segmented region is smaller than $t$, padding it with zeros to match the required size. We refer to these segmented heartbeat patches as individual ECG words, as they are independently extracted from each subject and lack cross-subject generalization.

**Generating ECG Sentences.** After segmentation, we concatenate the individual ECG words of the 12 leads in sequence, forming the overall ECG sentence $x \in \mathbb{R}^{l \times t}$, where $l$ represents the sequence length and $t$ denotes the time window size. Given the variability in heart rates across samples, the resulting sequence lengths are inconsistent. Similar to natural language processing, we set $l$ to the maximum length of the ECG sentence. If the length of the ECG sentence is less than $l$, it will be padded to $l$ through the zero-filled patches; if the length of the ECG sentence exceeds $l$, the interval length will be truncated to $l$. In this paper, $l$ is set to 256, and $t$ is set to 96.

## 3.2 ST-ECGFORMER BACKBONE NETWORK

To more effectively capture spatio-temporal features and latent semantic relationships within ECG sentences, we designed a backbone network called ST-ECGFormer. This backbone network is employed in various components of the HeartLang, including vector-quantized heartbeat reconstruction (VQ-HBR) training, masked ECG sentence pre-training, and downstream tasks fine-tuning.

**Token Embedding.** ECG signals exhibit high temporal resolution, and the QRS complexes that form ECG sentences contain rich temporal features. These QRS complexes are mapped into a higher-dimensional token feature space, allowing their distinguishing features to be more effectively extracted and encoded. We apply a 1-D convolutional layer-based mapping function to transform each individual ECG word into a corresponding token. After this transformation, the ECG sentence can be represented as $x_t \in \mathbb{R}^{l \times D}$, where $D$ denotes the dimension of the token feature space.

**Spatio-temporal and Position Embedding.** To enable the spatial and temporal information of the ECG sentence to be better captured by HeartLang, a temporal embedding set $TE = \{te_0, te_1, te_2, \ldots, te_{10}\}$ and a spatial embedding set $SE = \{se_0, se_1, se_2, \ldots, se_{12}\}$, both $D$-dimensional and learnable during the training process, were initialized. For the spatial embedding, we divide the original 12-lead ECG signals into 12 segments, with each lead corresponding to a spatial embedding. The spatial embedding of each individual ECG word is mapped back to the lead from which it originated. For the temporal embedding, the original 10-second ECG signal is divided into 10 segments, where each second corresponds to a temporal embedding. We assign the temporal embedding of each individual ECG word to the time interval of its QRS complex indices $Q$. Specifically, for zero-filled patches, their spatial and temporal embeddings are set to $te_0$ and $se_0$, respec-

tively, to ensure feature consistency. Next, a class token is added at the beginning of the sequence to enhance the representation. Additionally, a position embedding list $PE = \{pe_0, pe_1, pe_2, \ldots, pe_l\}$ is introduced to reinforce the sequential relationships between individual ECG words. Thus, the ECG sentence can be described by the following formula:

$$x' = x_t + TE_u + SE_v + PE,$$
$$u \in \{te_0, te_1, te_2, \ldots, te_{10}\}, v \in \{se_0, se_1, se_2, \ldots, se_{12}\}.$$

**Transformer Encoder.** Finally, the ECG sentence will be input into the transformer encoder (Vaswani et al., 2017). To ensure stability during the training process, we employ the pre-layer normalization strategy (Xiong et al., 2020), which applies layer normalization to the input of the attention mechanism:

$$Q = LN(x')w^Q, \quad K = LN(x')w^K, \quad V = LN(x')w^V,$$

$$\text{Attention}(Q, K, V) = \text{softmax}\left(\frac{QK^\top}{\sqrt{d_{head}}}\right)V,$$

where $d_{head}$ denotes the dimension of each head in the multi-head attention, and LN represents layer normalization.

### 3.3 VECTOR-QUANTIZED HEARTBEAT RECONSTRUCTION TRAINING

The individual ECG words segmented by the QRS-Tokenizer lack generalization properties, as each individual ECG word is independent. We aim for the HeartLang to learn general representations across subjects during the subsequent pre-training stage. To achieve this, we introduce an ECG vocabulary, a codebook containing collective ECG words that have cross-subject generalization properties. We believe that the same type of heartbeat across different subjects should be consistent in semantic level. Similar individual ECG words from different subjects are mapped to the same discrete and compact collective ECG word, allowing physiological differences between subjects to be overcome and form-level features of heartbeats to be learned. The construction of the ECG vocabulary is jointly optimized by quantization and reconstruction processes, as depicted in the upper half of Figure 2. This concept is inspired by VQ-NSP (Jiang et al., 2024), which encodes EEG signals into discrete latent representations and decodes them.

**Vector Quantization.** We first define an ECG vocabulary $\mathcal{V} = \{v_i | i = 1, \ldots, k\} \in \mathbb{R}^{k \times d}$, where $k$ is the number of collective ECG words in the vocabulary and $d$ is the dimension of each collective ECG word. Given an ECG signal sample $X \in \mathbb{R}^{C \times T}$, it is first generated by the QRS-Tokenizer into ECG sentence $x \in \mathbb{R}^{l \times t}$. After the ECG sentence is input into the ST-ECGFormer, a set of collective ECG word embeddings $P = \{p_i | i = 1, \ldots, l\}$ is obtained. Then, a quantizer is used to convert them into collective ECG word embeddings. The ECG vocabulary looks up the nearest neighbor of each interval representation $p_i$ in $V$. We use cosine similarity to find the closest collective ECG word embedding. This procedure can be formulated as

$$z_i = \arg\min \|\ell_2(p_i) - \ell_2(v_i)\|_2,$$

where $v_i$ is the collective ECG word embedding, and $\ell_2$ represents $\ell_2$ normalization.

**Heartbeat Reconstruction.** Due to the high signal-to-noise ratio of ECG signals, reconstructing the raw signals directly can efficiently train an ECG vocabulary and effectively learn form-level features of heartbeats. After being labeled by the quantizer, the normalized discrete collective ECG word embeddings $\{\ell_2(z_i) | i = 1, \ldots, l\}$ are fed into the transformer decoder. This process can be represented as

$$\hat{x} = \bigcup_{i=1}^{l} f_d\left(\ell_2(v_{z_i})\right),$$

where $\hat{x}$ is the reconstructed ECG sentence and $f_d$ is the decoder. To make the update of the ECG vocabulary more stable, we adopt an exponential moving average (EMA) strategy. The mean squared error (MSE) loss is utilized to guide the quantization and reconstruction processes. Finally, the loss function for training the VQ-HBR process is defined as

$$\mathcal{L}_V = \sum_{x \in \mathcal{D}} \sum_{i=1}^{l} \left( \begin{array}{c} \|\hat{x}_i - x\|_2^2 + \|sg(\ell_2(p_i)) - \ell_2(v_{z_i})\|_2^2 \\ + \|\ell_2(p_i) - sg(\ell_2(v_{z_i}))\|_2^2 \end{array} \right),$$

where $\mathcal{D}$ represents all the ECG sample data, and $sg$ denotes the stop-gradient operation, which is defined as the identity function in the forward pass and has zero gradient.

### 3.4 MASKED ECG SENTENCE PRE-TRAINING

The pre-training process of the HeartLang framework is illustrated in the lower part of Figure 2. In this stage, HeartLang learns generalized rhythm-level representations by masking parts of individual ECG words within an ECG sentence and predicting the corresponding collective ECG words based on the unmasked context. The pre-training stage primarily includes individual ECG words masking and collective ECG words prediction. This process is inspired by the work of Peng et al. (2022).

**Individual ECG Words Masking.** To enable the HeartLang to learn the rhythm-level features of ECG sentences, we perform random masking on the individual ECG words. This allows the HeartLang to understand the content of the entire ECG sentence based on the unmasked context. For the ECG sentence $x \in \mathbb{R}^{l \times t}$ obtained through the QRS-Tokenizer, where individual ECG words can be represented as: $e = \{e_i | i = 1, \dots, l\}$. We randomly generate a mask matrix $M = \{m_i | i = 1, \dots, l\}$, where $m_i \in \{0, 1\}$. Then, a learnable mask token $e_M \in \mathbb{R}^t$ is used to replace the masked heartbeats. Thus, the entire masked ECG sentence can be represented as $x_M = \{e_i : m_i = 0 | i = 1, \dots, N\} \cup \{e_M : m_i = 1 | i = 1, \dots, N\}$.

**Collective ECG Words Prediction.** The task of this stage is to predict the collective ECG word indices of the masked parts based on the unmasked individual ECG words, by minimizing the discrepancy between the predicted word indices and the true word indices. We extract the indices of the collective ECG words corresponding to the masked segments of the ECG sentence. The output hidden vectors are then denoted as $h = \{h_i \mid i = 1, \dots, l\}$, which are subsequently used to predict the corresponding collective ECG words via a linear classifier:

$$p(v'|e_M) = \text{softmax}(\text{Linear}(h)).$$

The target loss function for this stage is:

$$\mathcal{L}_P = -\sum_{x \in \mathcal{D}} \sum_{m_i = 1} \log p(v_i|e_M).$$

## 4 EXPERIMENTS

We follow the benchmark configuration and results provided by MERL (Liu et al., 2024). It is worth noting that the baseline methods in the benchmark were pre-trained on MIMIC-IV-ECG (Gow et al., 2023) dataset. Since MERL is a multimodal method based on ECG-Text, with an additional clinical report text supervision modality, while HeartLang performs pre-training and downstream fine-tuning solely on ECG recordings, we do not directly compare the two methods in the result presentation but instead compare with other eSSL methods.

### 4.1 VQ-HBR TRAINING AND PRE-TRAINING CONFIGURATION

**MIMIC-IV-ECG.** This publicly accessible dataset (Gow et al., 2023) contains 800,035 12-lead ECG recordings from 161,352 subjects. Each ECG recording was sampled at 500 Hz and lasted for 10 seconds. To prepare the pretraining dataset, we replaced the "NaN" and "Inf" values in the ECG recordings with the average of six neighboring points.

**Implementation.** Before VQ-HBR training and pre-training stage, we first downsampled all records in the dataset to 100 Hz and used the QRS-Tokenizer to transform the raw ECG recordings into a unified ECG sentence. We split the training and validation sets into 9:1, with the validation set data used for VQ-HBR training. In the VQ-HBR training stage, we set the learning rate to $5 \times 10^{-5}$ and trained for 100 epochs, with an ECG vocabulary size of 8,192 and a collective ECG word dimension of 128. In the pre-training stage, we set the learning rate to $5 \times 10^{-4}$, trained for 200 epochs, and applied a random masking rate of 50%. For both stages, a randomly initialized ST-ECGFormer was used as the backbone network, the AdamW optimizer was selected, and cosine annealing was applied for learning rate scheduling. All experiments were conducted on 8 NVIDIA GeForce RTX 4090 GPUs, with a batch size of 64 per GPU. We set the random seed to 0 to ensure the reproducibility of all results. More experimental details are provided in the *appendix*.

Table 1: Linear probing results of HeartLang and other eSSL methods. The best results are **bolded**, with gray indicating the second highest.

| Method | PTBXL-Super | | | PTBXL-Sub | | | PTBXL-Form | | | PTBXL-Rhythm | | | CPSC2018 | | | CSN | | |
|---|---|---|---|---|---|---|---|---|---|---|---|---|---|---|---|---|---|---|
| | 1% | 10% | 100% | 1% | 10% | 100% | 1% | 10% | 100% | 1% | 10% | 100% | 1% | 10% | 100% | 1% | 10% | 100% |
| SimCLR (Chen et al., 2020) | 63.41 | 69.77 | 73.53 | 60.84 | 68.27 | 73.39 | 54.98 | 56.97 | 62.52 | 51.41 | 69.44 | 77.73 | 59.78 | 68.52 | 76.54 | 59.02 | 67.26 | 73.20 |
| BYOL (Grill et al., 2020) | 71.70 | 73.83 | 76.45 | 57.16 | 67.44 | 71.64 | 48.73 | 61.63 | 70.82 | 41.99 | 74.40 | 77.17 | 60.88 | 74.42 | 78.75 | 54.20 | 71.92 | 74.69 |
| BarlowTwins (Zbontar et al., 2021) | 72.87 | 75.96 | 78.41 | 62.57 | 70.84 | 74.34 | 52.12 | 60.39 | 66.14 | 50.12 | 73.54 | 77.62 | 55.12 | 72.75 | 78.39 | 60.72 | 71.64 | 77.43 |
| MoCo-v3 (Chen et al., 2021) | 73.19 | 76.65 | 78.26 | 55.88 | 69.21 | 76.69 | 50.32 | 63.71 | 71.31 | 51.38 | 71.66 | 74.33 | 62.13 | 76.74 | 75.29 | 54.61 | 74.26 | 77.68 |
| SimSiam (Chen & He, 2021) | 73.15 | 72.70 | 75.63 | 62.52 | 69.31 | 76.38 | 55.16 | 62.91 | 71.31 | 49.30 | 69.47 | 75.92 | 58.35 | 72.89 | 75.31 | 58.25 | 68.61 | 77.41 |
| TS-TCC (Eldele et al., 2021) | 70.73 | 75.88 | 78.91 | 53.54 | 66.98 | 77.87 | 48.04 | 61.79 | 71.18 | 43.34 | 69.48 | 78.23 | 57.07 | 73.62 | 78.72 | 55.26 | 68.48 | 76.79 |
| CLOCS (Kiyasseh et al., 2021) | 68.94 | 73.36 | 76.31 | 57.94 | 72.55 | 76.24 | 51.97 | 57.96 | 72.65 | 47.19 | 71.88 | 76.31 | 59.59 | 77.78 | 77.49 | 54.38 | 71.93 | 76.13 |
| ASTCL (Wang et al., 2024) | 72.51 | 77.31 | 81.02 | 61.86 | 68.77 | 76.51 | 44.14 | 60.93 | 66.99 | 52.38 | 71.98 | 76.05 | 57.90 | 77.01 | 79.51 | 56.40 | 70.87 | 75.79 |
| CRT (Zhang et al., 2023c) | 69.68 | 78.24 | 77.24 | 61.98 | 70.82 | 78.67 | 46.41 | 59.49 | 68.73 | 47.44 | 73.52 | 74.41 | 58.01 | 76.43 | 82.03 | 56.21 | 73.70 | 78.80 |
| ST-MEM (Na et al., 2024) | 61.12 | 66.87 | 71.36 | 54.12 | 57.86 | 63.59 | 55.71 | 59.99 | 66.07 | 51.12 | 65.44 | 74.85 | 56.69 | 63.32 | 70.39 | 59.77 | 66.87 | 71.36 |
| **HeartLang (Ours)** | 78.94 | 85.59 | 87.52 | 64.68 | 79.34 | 88.91 | 58.70 | 63.99 | 80.23 | 62.08 | 76.22 | 90.34 | 60.44 | 66.26 | 77.87 | 57.94 | 68.93 | 82.49 |

## 4.2 Downstream Tasks Configuration

We evaluated our method on the three widely used public datasets listed below, which cover over 100 types of cardiac conditions. Detailed information on the data split can be found in the appendix.

**PTB-XL.** This publicly accessible dataset (Wagner et al., 2020) contains 21,837 12-lead ECG recordings collected from 18,885 patients. Each ECG recording was sampled at 500 Hz and lasted for 10 seconds. Based on the SCP-ECG protocol, the multi-class classification task has four subsets: **Superclass** (5 classes), **Subclass** (23 classes), **Form** (19 classes), and **Rhythm** (12 classes). We followed the official data split (Strodthoff et al., 2021) for training, validation, and testing.

**CPSC2018.** This publicly accessible dataset (Liu et al., 2018) contains 6,877 12-lead ECG recordings. Each recording was sampled at 500 Hz, with durations ranging from 6 to 60 seconds. The dataset is annotated with 9 different labels. We split the dataset into 70%:10%:20% for training, validation, and testing.

**Chapman-Shaoxing-Ningbo (CSN).** This publicly accessible dataset (Zheng et al., 2020; 2022) contains 45,152 12-lead ECG recordings. Each ECG recording was sampled at 500 Hz and lasted for 10 seconds. Following the configuration provided by MERL, we removed ECG records with "unknown" annotations. The refined version of the dataset contains 23,026 ECG recordings with 38 distinct labels. We split the dataset into 70%, 10%, 20% for training, validation, and testing.

**Implementation.** Before fine-tuning in downstream tasks, we first downsampled all records in the dataset to 100 Hz and used the QRS-Tokenizer to transform the raw ECG recordings into a unified ECG sentence. For linear probing, we kept the ST-ECGFormer backbone network frozen and only trained the randomly initialized parameters of the linear classifier. To explore the performance of our method under low-resource conditions, we conducted linear probing using 1%, 10%, and 100% of the training data for each task. We set the learning rate to $5 \times 10^{-3}$ and trained for 100 epochs. For the CPSC2018 and CSN datasets, we scaled the ECG recordings to the range of $[-3, 3]$ to enhance QRS detection. All test results were obtained from the best validation model, rather than testing the model on the test set after each epoch and reporting the highest result. For all downstream tasks, we used the macro AUC as the evaluation metric. We set the random seed to $0$ to ensure the reproducibility of all results. More experimental details are provided in the *appendix*.

## 5 Results and Discussions

### 5.1 Evaluation on Linear Probing

Table 1 presents the linear probing results of HeartLang compared to existing eSSL methods. In the PTB-XL dataset, HeartLang consistently demonstrated significant advantages across 1% to 100% of the training data. Specifically, compared to the second-best eSSL method, our method achieved an average macro AUC improvement of 8.14. Notably, on the Form and Rhythm subsets in PTB-XL dataset, HeartLang outperformed the second-best eSSL methods by an average of 9.85 in macro

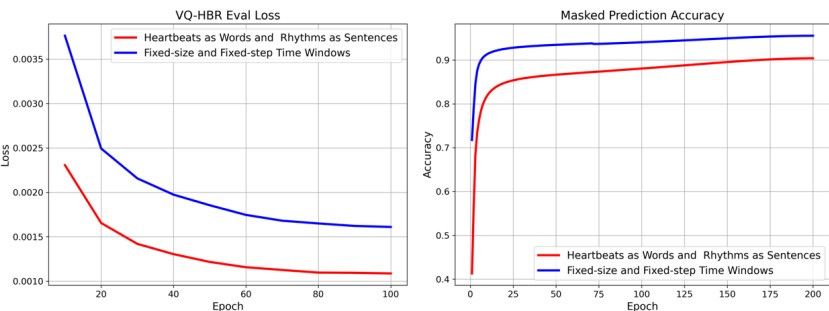

Figure 3: The validation loss curve during VQ-HBR training (left) and the prediction accuracy curve during masked ECG sentence pre-training (right), shown from two perspectives.

Table 2: Linear probing results of two signal slicing perspectives. The best results are **bolded**, while improved values are marked in green, and decreased values are marked in red.

| Perspective | PTBXL-Super | | | PTBXL-Sub | | | PTBXL-Form | | | PTBXL-Rhythm | | |
|---|---|---|---|---|---|---|---|---|---|---|---|---|
| | 1% | 10% | 100% | 1% | 10% | 100% | 1% | 10% | 100% | 1% | 10% | 100% |
| Fixed-size and Fixed-step Time Windows | 66.41 | 77.89 | 80.51 | 60.87 | 70.72 | 78.32 | **59.94** | **65.52** | 76.03 | 55.83 | 74.09 | 86.05 |
| Heartbeats as Words and Rhythms as Sentences | **78.94** | **85.59** | **87.52** | **64.68** | **79.34** | **88.91** | 58.70 | 63.99 | **80.23** | **62.08** | **76.22** | **90.34** |
| **Improvement** | 12.52 | 7.71 | 7.01 | 3.81 | 8.62 | 10.58 | -1.24 | -1.53 | 4.20 | 6.26 | 2.14 | 4.29 |

AUC with 100% of the training data, further highlighting significant advantages of HeartLang in ECG heartbeat and rhythm representation learning. This validates the effectiveness of our proposed signal slicing perspective of heartbeats as words and rhythms as sentences. For the CPSC2018 and CSN datasets, our method only outperformed others on the CSN dataset with 100% training data. We speculate that this is due to the significant baseline drift in these datasets, which may have reduced the performance of the QRS-Tokenizer. Nevertheless, our method remains highly competitive compared to other eSSL methods.

We attribute the weaker performance of other eSSL methods to their disruption of the semantic information in ECG signals. For contrastive eSSL methods, data augmentation methods such as rotation, shifting, and adding noise introduce semantic distortion to the positive and negative pairs in ECG signals, which leads to a decline in representation learning performance. For generative eSSL methods, treating ECG signals as ordinary time-series data and applying the fixed-size and fixed-step time windows for slicing cannot accommodate the broad and complex dynamic characteristics of ECG signals. This results in patches without clear semantic information, ultimately leading to a decline in representation learning performance. In contrast, our method uses the QRS-Tokenizer to segment ECG patches with clear semantic meaning and enhances representation learning by reconstructing both heartbeat form and cardiac rhythms, ultimately achieving superior performance in downstream tasks.

## 5.2 EVALUATION ON SIGNAL SLICING PERSPECTIVE

To further validate the effectiveness of our proposed "heartbeats as words, rhythms as sentences" signal slicing perspective, we compared it with the traditional signal slicing perspective in this experiment, which utilizes the fixed window size and time step. For fixed-size and fixed-step time windows perspective, we created ECG sentences from the raw ECG recordings in the same manner but without the QRS detection process, instead slicing based on fixed window sizes and strides. We then performed VQ-HBR training, pre-training, and linear probing using the same configurations.

Figure 3 presents the VQ-HBR training loss curves and masked ECG sentence pre-training prediction accuracy curves for both perspectives. During the VQ-HBR training stage, our proposed perspective significantly outperforms the traditional signal slicing perspective. This is because the

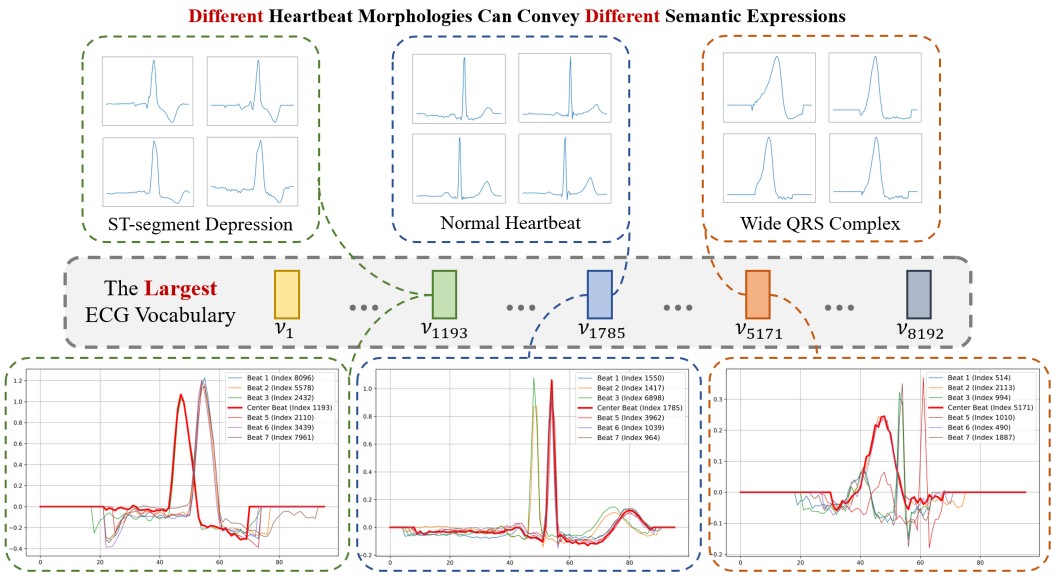

Figure 4: ECG vocabulary visualization.

patches generated by our method carry clear semantic information, which benefits the training of the ECG vocabulary. However, in the pre-training stage, the traditional slicing perspective shows higher masked prediction accuracy. We attribute this difference primarily to the number of zero-filled patches. The ECG sentences generated by the traditional slicing perspective use a fixed 120 patches, resulting in a larger number of zero-filled patches. In contrast, the number of patches used in our proposed perspective is dynamic and dependent on the heart rate. The larger number of zero-filled patches in the traditional perspective may improve the performance of masked prediction during pre-training. Table 2 presents the results of downstream fine-tuning on the PTB-XL dataset using two different perspectives. In the majority of cases, our perspective resulted in significant improvements in macro AUC performance, except for a few low-resource scenarios where there was a slight performance drop. Specifically, our perspective resulted in an average improvement of 5.36 in macro AUC. Notably, on the Superclass and Subclass subset, our perspective brought an average improvement of 8.38 in macro AUC.

Despite the fact that masked ECG sentence pre-training prediction accuracy of our perspective was lower than that of the traditional perspective during the pre-training stage, it achieved better results in downstream tasks. We attribute this to the higher modeling difficulty in masked ECG sentence pre-training for our perspective, which helped the model uncover latent semantic relationships between heartbeats, leading to the learning of more generalized representations. Interestingly, in the Superclass and Subclass subsets, using the traditional perspective caused our method to degrade to baseline-like performance. We speculate that if baseline methods adopted our perspective, they would also see performance improvements. Furthermore, in the Form and Rhythm subsets, even with the traditional perspective, our method outperformed baseline methods, further demonstrating the effectiveness of the HeartLang architecture in multi-level representation learning of ECG signals.

## 5.3 ECG VOCABULARY VISUALIZATION

In this section, we visualize the ECG vocabulary to reflect how the collective ECG words correspond to the original individual ECG words. We trained and validated VQ-HBR on MIMIC-IV-ECG, where the effective use of collective ECG words in the validation set amounted to 5,394 (discrete words). In Figure 4, the individual ECG words with the same index exhibit similar and clear semantic information in terms of heartbeat representation. For instance, index 1193 displays the morphological features of ST-segment depression, while index 5171 shows the characteristics of a wide QRS complex. We further visualized ECG sentence segments centered around individual ECG

Table 3: Linear probing results of the ablation study. The best results are **bolded**, with gray indicating the second highest.

| Method | PTBXL-Super | | | PTBXL-Sub | | | PTBXL-Form | | | PTBXL-Rhythm | | |
|---|---|---|---|---|---|---|---|---|---|---|---|---|
| | 1% | 10% | 100% | 1% | 10% | 100% | 1% | 10% | 100% | 1% | 10% | 100% |
| w/o ECG Vocabulary | 59.23 | 61.51 | 79.68 | 49.34 | 50.55 | 73.16 | 49.33 | 46.88 | 63.25 | 42.87 | 43.87 | 77.65 |
| w/o Pre-training | 70.34 | 77.82 | 80.31 | 55.70 | 67.64 | 80.57 | 53.73 | 57.67 | 66.70 | **65.98** | 76.49 | 81.40 |
| w/o Spatio-temporal Embedding | 69.78 | 81.79 | 85.12 | 58.06 | 73.76 | 87.33 | 55.09 | 63.37 | 73.90 | 61.79 | 74.83 | 84.93 |
| w/o Spatial Embedding | 78.74 | 85.32 | 86.87 | **67.82** | 79.31 | 88.66 | 60.76 | **67.66** | 79.10 | 63.54 | 79.00 | 89.25 |
| w/o Temporal Embedding | 77.87 | 84.86 | 85.85 | 64.56 | 77.48 | 87.50 | **61.41** | 67.54 | 77.50 | 65.44 | **84.74** | 87.81 |
| **HeartLang (Ours)** | **78.94** | **85.59** | **87.52** | 64.68 | **79.34** | **88.91** | 58.70 | 63.99 | **80.23** | 62.08 | 76.22 | **90.34** |

words at three distinct indices, each surrounded by six neighboring individual ECG words (three on each side). In our constructed ECG sentences, the incorporation of spatio-temporal and position embeddings leads to similar heartbeats (e.g., normal heartbeat) within the same ECG sentence being mapped to different collective ECG words. This phenomenon suggests that collective ECG words encapsulate spatio-temporal and positional features, providing richer semantic information. Similarly, in English, the same word can serve different grammatical functions (e.g., 'run' can be a verb or a noun). The same words (similar heartbeat morphology) with identical spellings can assume different parts of speech (different collective ECG words) depending on the context (spatio-temporal and position embeddings). However, in previous ELP studies, vocabularies were only constructed by clustering morphologically similar ECG waves for modeling (not heartbeats, no more than 70 clusters), without incorporating spatio-temporal and positional information, which resulted in limited parts of speech. Expressions of different parts of speech through the context further highlights the richer semantic representation of our study compared to previous ELP studies.

## 5.4 ABLATION STUDY

In this section, we present a comprehensive ablation study of HeartLang to illustrate the effectiveness of each component within the framework, as shown in Table 3. For the experiment without the ECG vocabulary, we discarded it and applied mean squared error (MSE) loss to reconstruct the masked segments. This approach did not allow HeartLang to effectively learn general representations during pre-training, highlighting the role of ECG vocabulary in helping HeartLang capture latent semantic relationships in ECG sentences. For the experiment without pre-training, we fine-tuned a randomly initialized ST-ECGFormer on downstream tasks. The results indicated that the pretraining process effectively learned general representations in ECG sentences, leading to significant improvements in downstream performance. We also conducted an ablation study on the structure of ST-ECGFormer, completing all training and fine-tuning procedures. The results showed that, aside from a slight performance drop under low-resource conditions, HeartLang consistently enhanced performance, demonstrating that ST-ECGFormer effectively learns spatio-temporal representations of ECG signals and boosts the performance of HeartLang on downstream tasks.

## 6 CONCLUSION

In this paper, we propose a novel perspective on ECG signal processing, treating them as a language with distinct words (heartbeats) and sentences (rhythms). Based on this perspective, we introduce the QRS-Tokenizer, which generates the ECG sentences from the raw ECG signals. Building upon these, we propose HeartLang, a novel self-supervised learning framework for ECG language processing. HeartLang learns form-level general representations through vector-quantized heartbeat reconstruction training and rhythm-level general representations through masked ECG sentence pre-training. Additionally, we constructed the largest heartbeat-based ECG vocabulary to date. This ECG vocabulary includes a wide variety of heartbeat morphological representations across different cardiac condition, which will further advance the development of ECG language processing. We evaluated HeartLang across six public ECG datasets, where it demonstrated robust competitiveness against other eSSL methods. We hope that the ideas presented here can inspire the ECG research community, particularly in the emerging field of ECG language processing.

## ACKNOWLEDGMENTS

This work was supported by the National Natural Science Foundation of China (62102008, 62172018); CCF-Zhipu Large Model Innovation Fund (CCF-Zhipu202414); STI 2030-Major Projects (2022ZD0208900); and the Guangdong Basic and Applied Basic Research Foundation (2024A1515010524).

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

# A DETAIL IN EXPERIMENTAL SETTINGS

## A.1 HYPERPARAMETER SETTINGS

In this section, we provide detailed hyperparameter settings for HeartLang. Due to computational resource limitations, the VQ-HBR training and pre-training for the ablation experiments were conducted using four NVIDIA GeForce RTX 4090 GPUs, with a batch size of 64 per GPU.

Table 4: Hyperparameters for vector-quantized heartbeat reconstruction training.

| Hyperparameters | Values |
| --- | --- |
| Transformer encoder layers | 4 |
| Transformer decoder layers | 2 |
| Hidden size | 768 |
| MLP size | 2048 |
| Attention head number | 2 |
| Vocabulary size | 8192 |
| Collective ECG word dimension | 128 |
| Batch size | 512 |
| Peak learning rate | 5e-5 |
| Minimal learning rate | 1e-5 |
| Learning rate scheduler | Cosine |
| Optimizer | AdamW |
| Adam $\beta$ | (0.9,0.99) |
| Weight decay | 1e-4 |
| Total epochs | 100 |
| Warmup epochs | 10 |

Table 5: Hyperparameters for masked ECG sentence pre-training.

| Hyperparameters | Values |
| --- | --- |
| Transformer encoder layers | 12 |
| Hidden size | 768 |
| MLP size | 1024 |
| Attention head number | 8 |
| Batch size | 512 |
| Peak learning rate | 5e-4 |
| Minimal learning rate | 1e-5 |
| Learning rate scheduler | Cosine |
| Optimizer | AdamW |
| Adam $\beta$ | (0.9,0.98) |
| Weight decay | 0.05 |
| Total epochs | 200 |
| Warmup epochs | 5 |
| Gradient clipping | 3.0 |
| Mask ratio | 0.5 |

Table 6: Hyperparameters for downstream fine-tuning.

| Hyperparameters | Values |
|---|---|
| Batch size | 256 |
| Peak learning rate | 5e-3 |
| Minimal learning rate | 1e-5 |
| Learning rate scheduler | Cosine |
| Optimizer | AdamW |
| Weight decay | 0.05 |
| Layer decay | 0.9 |
| Total epochs | 100 |
| Warmup epochs | 10 |

## A.2 DATASET SPLIT

We describe the dataset split in Table 7. For the MIMIC-IV-ECG (Gow et al., 2023) dataset, we split the training and validation sets with a ratio of 9:1. The validation set of this dataset is used during the VQ-HBR training stage, but not in the pretraining stage. For the four subsets of PTBXL, we adhere to the official split from the official work (Wagner et al., 2020). For CPSC2018 (Liu et al., 2018) and CSN (Zheng et al., 2020; 2022), We followed the division method provided by MERL (Liu et al., 2024) to ensure consistency.

Table 7: Details on dataset split.

| Dataset | Number of Categories | Train | Valid | Test |
|---|---|---|---|---|
| MIMIC-IV-ECG (Gow et al., 2023) | - | 720,031 | 80,004 | - |
| PTBXL-Super (Wagner et al., 2020) | 5 | 17,084 | 2,146 | 2,158 |
| PTBXL-Sub (Wagner et al., 2020) | 23 | 17,084 | 2,146 | 2,158 |
| PTBXL-Form (Wagner et al., 2020) | 19 | 7,197 | 901 | 880 |
| PTBXL-Rhythm (Wagner et al., 2020) | 12 | 16,832 | 2,100 | 2,098 |
| CPSC2018 (Liu et al., 2018) | 9 | 4,950 | 551 | 1,376 |
| CSN (Zheng et al., 2020; 2022) | 38 | 16,546 | 1,860 | 4,620 |

# B    MORE RESULTS AND DISCUSSIONS

## B.1    EVALUATION ON DIFFERENT VOCABULARY SIZES

We present the performance of downstream tasks under different vocabulary sizes in Table 8. We reduced the vocabulary size to 64, similar to the vocabulary size used in previous ECG language processing studies, thereby limiting its semantic expressions. The results show that a larger vocabulary size leads to significant performance improvements. A larger vocabulary provides richer semantic representations and increases the complexity of pre-training, enabling the model to learn more generalized representations and improve the performance of downstream tasks.

Table 8: Linear probing results of two different vocabulary sizes. The best results are **bolded**, while improved values are marked in **green**.

| Vocabulary Size | PTBXL-Super | | | PTBXL-Sub | | | PTBXL-Form | | | PTBXL-Rhythm | | |
|---|---|---|---|---|---|---|---|---|---|---|---|---|
| | 1% | 10% | 100% | 1% | 10% | 100% | 1% | 10% | 100% | 1% | 10% | 100% |
| 64 | 75.89 | 83.72 | 86.23 | 60.97 | 75.95 | 86.88 | 57.10 | 62.99 | 75.69 | 58.41 | 75.31 | 88.51 |
| 8192 | **78.94** | **85.59** | **87.52** | **64.68** | **79.34** | **88.91** | **58.70** | **63.99** | **80.23** | **62.08** | **76.22** | **90.34** |
| **Improvement** | **3.05** | **1.87** | **1.29** | **3.71** | **3.39** | **2.03** | **1.60** | **1.00** | **4.54** | **3.67** | **0.91** | **1.83** |

## B.2    EVALUATION ON FEWER LEAD CONFIGURATION

Based on the fewer lead configuration recommended by Oh et al. (2022), the configuration is shown in Table 9, and the downstream task validation results are presented in Table 10. In most cases, the downstream performance improves significantly with the increase in the number of leads, especially in the Superclass and Subclass subsets for disease diagnosis. Notably, even under the single-lead condition, the downstream task performance of HeartLang surpasses that of most baseline methods in Table 1. This demonstrates that HeartLang is well-adapted to the special case of single-lead configurations, highlighting its strong generalization capability.

Table 9: Fewer lead configuration and selected leads.

| Number of Leads | Selected Leads |
|---|---|
| 1 | I |
| 2 | I, II |
| 3 | I, II, V2 |
| 6 | I, II, III, aVR, aVL, aVF |
| 12 | I, II, III, aVR, aVL, aVF, V1, V2, V3, V4, V5, V6 |

Table 10: Linear probing results of five different lead combinations. The best results are **bolded**, with gray indicating the second highest.

| Number of Leads | PTBXL-Super | | | PTBXL-Sub | | | PTBXL-Form | | | PTBXL-Rhythm | | |
|---|---|---|---|---|---|---|---|---|---|---|---|---|
| | 1% | 10% | 100% | 1% | 10% | 100% | 1% | 10% | 100% | 1% | 10% | 100% |
| 1 | 73.97 | 79.74 | 81.02 | 66.91 | 77.04 | 83.11 | 58.66 | 66.06 | 70.98 | 55.28 | 74.53 | 84.53 |
| 2 | 76.81 | 83.70 | 85.14 | **69.63** | 79.12 | 85.89 | 59.42 | **68.27** | 77.16 | 61.57 | 81.60 | 86.98 |
| 3 | 76.55 | 84.12 | 85.97 | 66.61 | 78.26 | 87.68 | 55.47 | 67.76 | 70.46 | **68.19** | 83.47 | 86.27 |
| 6 | 76.45 | 83.72 | 85.66 | 62.59 | 77.52 | 85.92 | **59.74** | 68.03 | 79.46 | 63.60 | **83.80** | **91.44** |
| 12 | **78.94** | **85.59** | **87.52** | 64.68 | **79.34** | **88.91** | 58.70 | 63.99 | **80.23** | 62.08 | 76.22 | 90.34 |

### B.3 Limitations and Future Works

Our proposed QRS-Tokenizer relies on the QRS complex features of heartbeats, which are a critical component of its functionality. However, for certain diseases or conditions characterized by irregular QRS complexes, the tokenizer may struggle to accurately represent these atypical patterns, leading to performance degradation. Additionally, when segmenting heartbeats, the QRS-Tokenizer pads intervals smaller than 96 with zeros. While this approach is simple and effective for most high-quality datasets, it can partially disrupt the characteristics of heartbeats in datasets with significant baseline drift, as their baselines may deviate substantially from zero. These challenges highlight the need for future improvements to the QRS-Tokenizer, with a focus on enhancing its robustness to handle both irregular QRS complexes and baseline drift effectively, paving the way for more reliable ECG language processing across diverse cardiac conditions.

# C  MORE VISUALIZATION RESULTS

## C.1  INDIVIDUAL ECG WORDS AND COLLECTIVE ECG WORDS VISUALIZATION

We further visualized additional individual ECG words and collective ECG words to demonstrate the semantic richness of our constructed ECG vocabulary, as shown in Figure 5.

## C.2  ECG SENTENCE VISUALIZATION

We visualize the constructed ECG Sentences, as illustrated in Figures 6, 7, and 8. In these figures, the blue lines correspond to the original signals, while the red lines denote the reconstructed signals. As shown in Figure 6, the reconstructed signal demonstrates a notably smoother profile, suggesting that the collective ECG word has effectively captured the morphological characteristics of the heartbeat and exhibits strong generalization capabilities. One notable feature of the QRS-Tokenizer we designed is its ability to adaptively segment individual ECG words based on heart rate, as demonstrated in Figures 7 and 8. In Figure 7, due to the relatively fast heart rate, the individual ECG words within the ECG Sentence utilize the full sentence length of 256. In contrast, Figure 8 shows a relatively slower heart rate, resulting in the use of only a smaller portion of the ECG Sentence, with the remaining sections zero-filled.

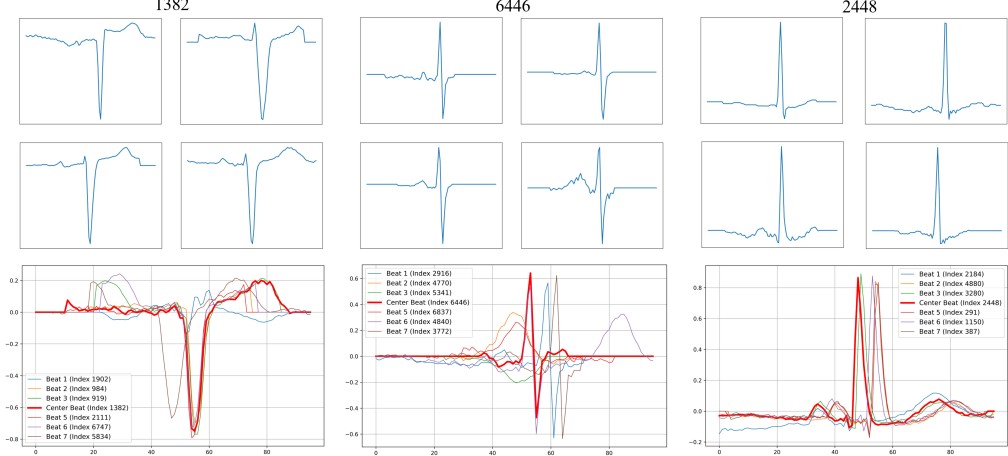

Figure 5: More ECG Words Visualization.

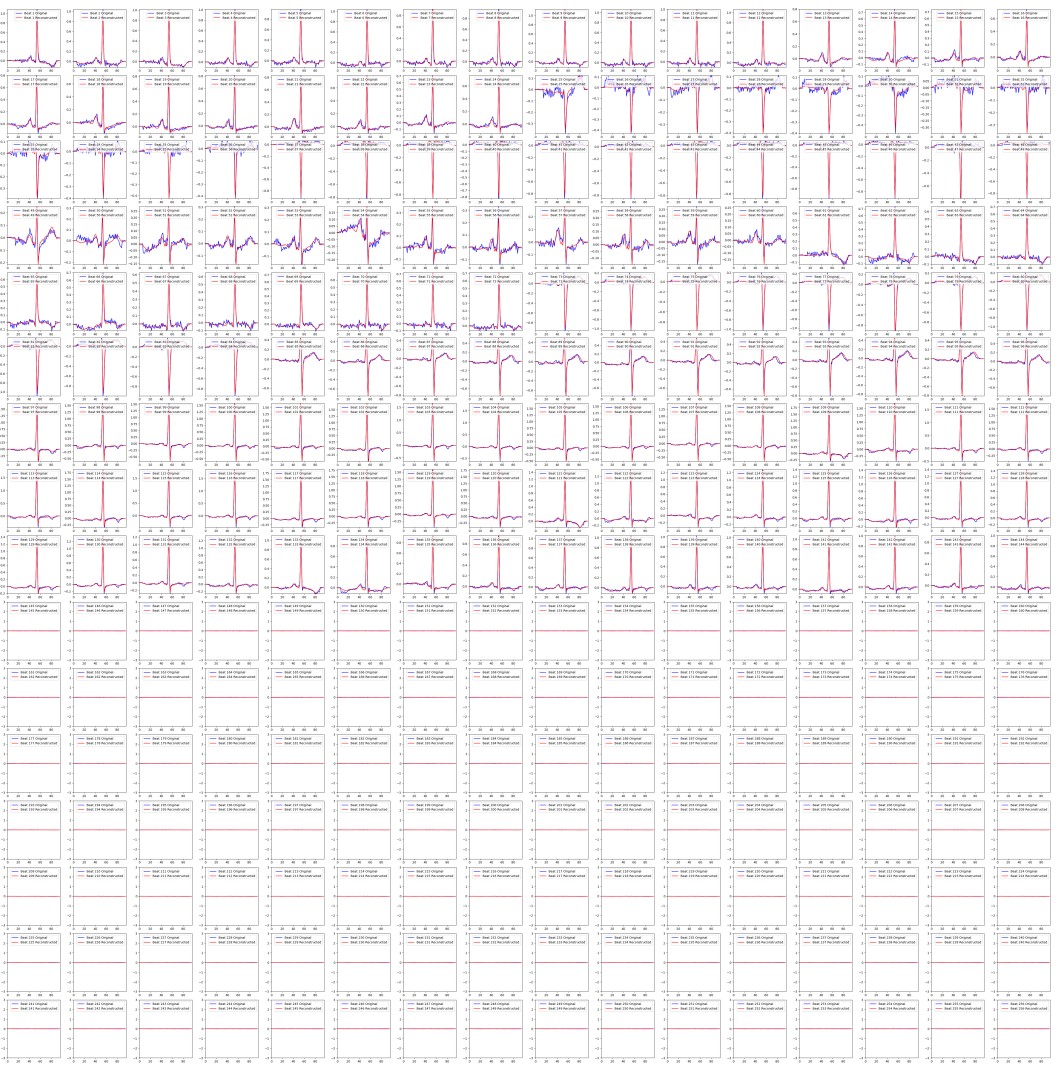

Figure 6: ECG Sentence Visualization.

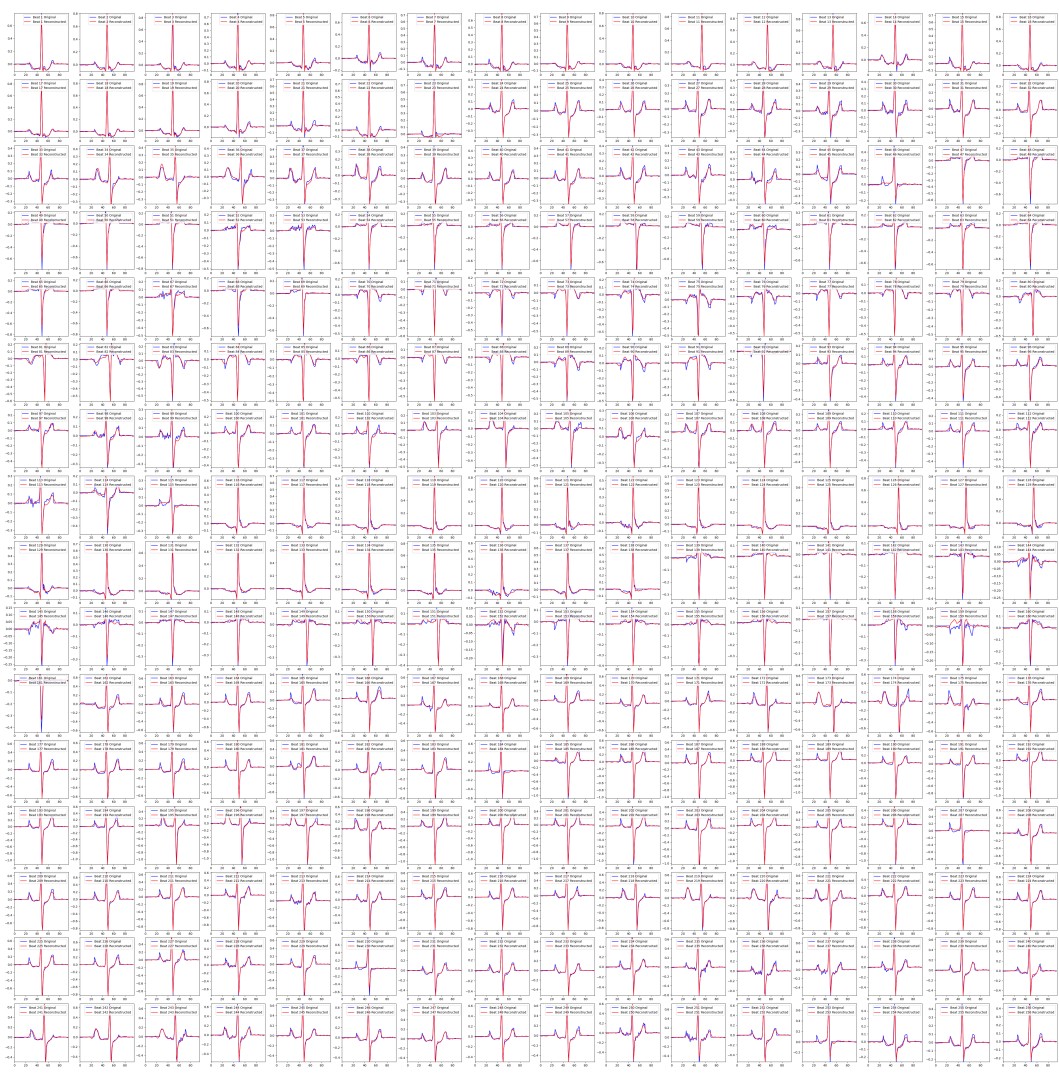

Figure 7: Visualization of the ECG Sentence with Fast Heart Rate.

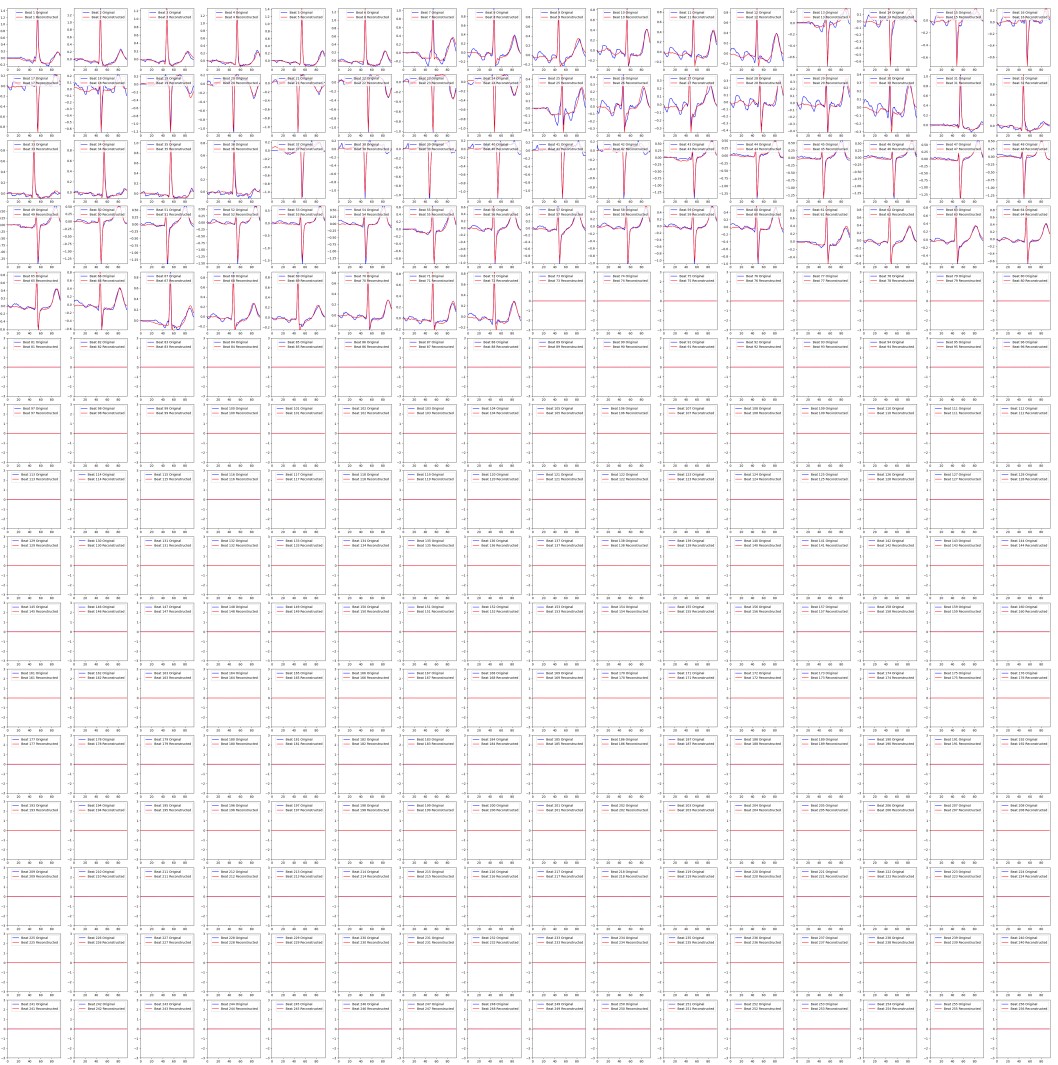

Figure 8: Visualization of the ECG Sentence with Slow Heart Rate.

