# OpenReview forum: "Reading Your Heart: Learning ECG Words and Sentences via Pre-training ECG Language Model"
_ICLR.cc/2025/Conference — ICLR 2025 Poster_

### Official Review · Reviewer_xUd7 · 2024-10-26

**Soundness:** 3
**Presentation:** 4
**Contribution:** 3
**Rating:** 6
**Confidence:** 4

**Summary:**

This paper proposes a self-supervised learning framework for ECG language processing considering ECG beats as words and sequence of of beats as sentences mimicking learning ECG representation at form and rhythm levels. Instead of segmenting ECG signal into fixed-size and fixed-step windows view, as done by other contemporary studies, in this study, ECG beats (area consisting of PQSRT waves) are isolated by locating R-peaks and area around it and repeating it to retrieve sequence of beats for analysis. Transformer-based backbone architecture was used as encoder named as ST-ECGFormer to input ECG sentences which consists of ECG word (token) embedding and spatio-temporal and position embedding. Using vector quantisation, ECG vocabulary was auto generated in order to identify and cross-match similar ECG beats (words). A reconstruction training was adopted by masking parts of ECG words and using encoder/decoder architecture to predict the collective ECG word indices of the masked parts based on the unmasked individual ECG words. Authors have done experimentation on 3 publicly available ECG datasets and the proposed method achieved superior performance on heartbeat class and rhythm class classification downstream tasks after pre-training followed by fine-tuning which shows superior performance in several cases. ECG vocabulary was created and visualisation shows ECG words with the same index exhibit similar semantic information in terms of heartbeat representation.

**Strengths:**

A self-supervised framework was proposed for ECG representation learning considering an ECG signals as a sequence of beats, thus slicing them to include each of them in order to learn their form and a sequence of them to learn rhythms. A ECG vocabulary was created  which shows similar semantic information in terms of heartbeat representation.

**Weaknesses:**

The study lacks novelty in the framework itself since it is based on an existing one published in past ICLR venue, titled "LARGE BRAIN MODEL FOR LEARNING GENERIC REPRESENTATIONS WITH TREMENDOUS EEG DATA IN  BCI". However, the novelty lies in considering another modality of data, ECG. The way ECG beats are curated to formulate the problem in this study is interesting.

- The performance difference compared to other self-supervised methods in the literature were discussed, however, it is unclear if the difference is due to the proposed framework or slicing ECG precisely to consider each beat in forming a sequence of beats, or just for considering ECG segments fixed-size and fixed-step. Applying the beat slicing mechanism used in this study to another study where fixed-size and fixed-step windowing were used would help.

- The generated ECG vocabulary is interesting to see, however, its motivation is unclear considering the fact that normal beats of a single recording may yield to show variance in the vocabulary without any clinical significance. It is not shown in this study how diverse these beats are for each category such as how many words the model comes up with for normal beats of a single recording. A domain specific discussion seems necessary to understand the utility of ECG vocabulary.

**Questions:**

Overall, the paper could be a moderate ECG representation learning contribution, with some practical and experimental issues requiring clarification. Given these clarifications in an author response, I would be willing to increase the score.

The ECG vocabulary was learned which shows similar semantic information, however, its consistency, as well as its randomness needs to be rigorously explored. It's necessity seems questionable due to the fact that normal heartbeats from a single recording may have turned into separate ECG words in the vocabulary which does not make much sense. Although the methodological process can generate it, its utility needs to be justified first. This thus undermines one of the claimed novelty.  An analysis of vocabulary stability across different recordings or subjects seems necessary.

12-lead ECGs were considered as data where some leads can be directly deduced from lead II and III including I, aVR, aVL, and aVF - such channel redundancies may have impact on ECG vocabulary generation. The framework does not show generalisability for less number of channels or even a single channel ECG where there is no spatial items. Experiments with reduced lead sets or single-channel ECGs would be beneficial that would demonstrate the framework's generalizability.

Data split lacks clarity if the train, test and validation splits combine segments across recordings or subject wise splits were considered. A clarification of data splitting strategy in the methodology seems necessary. This is because, if subject wise splits were not considered, then there will be data leakage.

The proposed framework is a variant of EEG based framework, however, the contribution is apparently for ECG modality and the data slicing philosophy. Authors should clarify the claim of framework based novelty accordingly.

---

> ### Author Response · Authors · 2024-11-19
> **Response to Reviewer xUd7 (1/3)**
>
> We are extremely grateful for your review of the manuscript. You have raised a number of important issues. We agree with your comments and have modified our manuscript accordingly. Below we give a point-by-point response to your concerns and suggestions.
>
> > **Q1: The ECG vocabulary was learned which shows similar semantic information, however, its consistency, as well as its randomness needs to be rigorously explored. It's necessity seems questionable due to the fact that normal heartbeats from a single recording may have turned into separate ECG words in the vocabulary which does not make much sense. Although the methodological process can generate it, its utility needs to be justified first. This thus undermines one of the claimed novelty. An analysis of vocabulary stability across different recordings or subjects seems necessary.**
>
> **A1:** We would first like to clarify why similar normal heartbeats within the same recording may be mapped to different ECG words. For similar normal heartbeats within a single ECG recording, due to the effects of Spatio-temporal Embedding and Position Embedding, normal heartbeats at different lead positions, different time points, and different sequence positions generate distinct embeddings. These different embeddings are then mapped to the nearest vector in the ECG vocabulary (i.e., different collective ECG words). The entire vector-quantized heartbeat reconstruction process dynamically guides the construction of the ECG vocabulary through both quantization loss and reconstruction loss. In this process, Spatio-temporal Embedding and Position Embedding provide contextual information, influencing the embedding of normal heartbeats before mapping, ultimately resulting in mappings to different collective ECG words.
>
> In previous natural language processing research, dynamic word embeddings that incorporate contextual information have been shown to be superior to static word embedding methods [1]. Dynamic embeddings are now the standard processing paradigm in NLP (e.g., BERT, T5, GPT).
>
> Our study adopts a similar approach, which may seem counterintuitive. However, we believe that context-rich ECG words are meaningful, as they lead to more semantically rich representations. In self-supervised pretraining, where there is no additional supervision, the model learns directly from the data itself. Richer semantic expressions increase the complexity of the pretraining process, allowing the model to learn more generalized representations. We conducted additional experiments on PTB-XL datasets with an ECG vocabulary size of 64, which limits the semantic expressions of ECG words and is similar to the vocabulary size used in previous ECG language processing studies. The downstream validation results are presented in the table below.
>
> | Vocabulary Size |          |  Super   |          |          |   Sub    |          |          |   Form   |          |          |  Rhythm  |          |
> | :-------------: | :------: | :------: | :------: | :------: | :------: | :------: | :------: | :------: | :------: | :------: | :------: | :------: |
> |                 |    1%    |   10%    |   100%   |    1%    |   10%    |   100%   |    1%    |   10%    |   100%   |    1%    |   10%    |   100%   |
> |       64        |  75.89   |  83.72   |  86.23   |  60.97   |  75.95   |  86.88   |  57.10   |  62.99   |  75.69   |  58.41   |  75.31   |  88.51   |
> |      8192       |  78.94   |  85.59   |  87.52   |  64.68   |  79.34   |  88.91   |  58.70   |  63.99   |  80.23   |  62.08   |  76.22   |  90.34   |
> | **Improvement** | **3.05** | **1.87** | **1.29** | **3.71** | **3.39** | **2.03** | **1.60** | **1.00** | **4.54** | **3.67** | **0.91** | **1.83** |
>
> The results show that as the vocabulary size increases, the performance of downstream tasks improves significantly. This indicates that enhanced semantic expressions lead to better representations learned during the pretraining stage. Due to the influence of contextual information, even similar normal heartbeats are mapped to different collective ECG words, enriching the semantic expressions. This, in turn, enables the model to learn more general representations during pretraining, ultimately boosting the performance of downstream tasks. In summary, a more diverse collection of ECG words will result in richer semantic expressions, which in turn will enhance the performance of both pre-training and downstream tasks. The relevant results and discussion have been added to the appendix.

---

> ### Author Response · Authors · 2024-11-19
> **Response to Reviewer xUd7 (2/3)**
>
> > **Q2: 12-lead ECGs were considered as data where some leads can be directly deduced from lead II and III including I, aVR, aVL, and aVF - such channel redundancies may have impact on ECG vocabulary generation. The framework does not show generalisability for less number of channels or even a single channel ECG where there is no spatial items. Experiments with reduced lead sets or single-channel ECGs would be beneficial that would demonstrate the framework's generalizability.**
>
> **A2:** Although these channels may seem redundant, as mentioned in the response to Question 1, larger ECG words enhance semantic expressions, thereby improving both pretraining and downstream task performance. Therefore, we chose to train the model using the standard 12-lead ECG configuration, which is also commonly used in clinical diagnostics. Regarding the adaptability to fewer leads or single-lead ECGs, we conducted additional experiments based on the configuration in [2], and the results are presented in the table below.
>
> | Number of Leads |           |   Super   |           |           |    Sub    |           |           |   Form    |           |           |  Rhythm   |           |
> | :-------------: | :-------: | :-------: | :-------: | :-------: | :-------: | :-------: | :-------: | :-------: | :-------: | :-------: | :-------: | :-------: |
> |                 |    1%     |    10%    |   100%    |    1%     |    10%    |   100%    |    1%     |    10%    |   100%    |    1%     |    10%    |   100%    |
> |        1        |   73.97   |   79.74   |   81.02   |   66.91   |   77.04   |   83.11   |   58.66   |   66.06   |   70.98   |   55.28   |   74.53   |   84.53   |
> |        2        |   76.81   |   83.70   |   85.14   | **69.63** |   79.12   |   85.89   |   59.42   | **68.27** |   77.16   |   61.57   |   81.60   |   86.98   |
> |        3        |   76.55   |   84.12   |   85.97   |   66.61   |   78.26   |   87.68   |   55.47   |   67.76   |   70.46   | **68.19** |   83.47   |   86.27   |
> |        6        |   76.45   |   83.72   |   85.66   |   62.59   |   77.52   |   85.92   | **59.74** |   68.03   |   79.46   |   63.60   | **83.80** | **91.44** |
> |       12        | **78.94** | **85.59** | **87.52** |   64.68   | **79.34** | **88.91** |   58.70   |   63.99   | **80.23** |   62.08   |   76.22   |   90.34   |
>
> The results demonstrate that downstream task performance generally improves with an increasing number of leads, particularly in the Superclass and Subclass subsets for disease diagnosis. Remarkably, even under the single-lead condition, HeartLang outperforms most baseline methods reported in Table 1. This highlights exceptional adaptability of HeartLang  to single-lead configurations and underscores its robust generalization capability across different lead configurations. The related results and discussions have been supplemented in the appendix.
>
> > **Q3: Data split lacks clarity if the train, test and validation splits combine segments across recordings or subject wise splits were considered. A clarification of data splitting strategy in the methodology seems necessary. This is because, if subject wise splits were not considered, then there will be data leakage.**
>
> **A3:** For the division of datasets, we strictly follow the method recommended by MERL (ICML 2024) to ensure fairness in comparing downstream validation results. Specifically, for the PTB-XL series datasets, we use the official dataset processing code for partitioning (https://github.com/helme/ecg_ptbxl_benchmarking). As stated in the PTB-XL original paper, the data division adheres to the principle of patient independence, ensuring that ECG data from the same patient does not appear in different folds, thereby avoiding data leakage. For the CPSC2018 and CSN datasets, we utilize the CSV files provided by MERL for partitioning (https://github.com/cheliu-computation/MERL-ICML2024/tree/main/finetune/data_split). Although the MERL paper does not explicitly illustrate the issue of patient independence, to ensure fairness in comparisons with the baseline methods, we have adopted this approach for dataset partitioning.

---

> ### Author Response · Authors · 2024-11-19
> **Response to Reviewer xUd7 (3/3)**
>
> > **Q4: The proposed framework is a variant of EEG based framework, however, the contribution is apparently for ECG modality and the data slicing philosophy. Authors should clarify the claim of framework based novelty accordingly.**
>
> **A4:** Here are the architectural differences between our approach and LaBraM:
>
> - **QRS-Tokenizer for Implementing the New Perspective:** The QRS-Tokenizer is a key component for realizing the concept of "Heartbeats as Words and Rhythms as Sentences," which significantly sets our work apart from previous studies on physiological signals. Unlike the Neural Tokenizer in LaBraM, which essentially follows the traditional slicing approach with fixed-size and fixed-step time windows and lacks the semantic concept, our approach introduces a novel perspective. As highlighted in Section 5.2, "Evaluation on Signal Slicing Perspective," the "Heartbeats as Words and Rhythms as Sentences" perspective achieves an average macro AUC improvement of 5.36 compared to the "Fixed-size and Fixed-step Time Windows" approach. Notably, for superclass and subclass subsets, our perspective yields an average macro AUC increase of 8.38. These improvements underscore the innovation of our proposed perspective and the QRS-Tokenizer.
> - **Reconstruction objectives during the vocabulary construction stage:** In LaBraM, the reconstruction objective for its VQ-NSP process is the Fourier Spectrum of EEG signal patches. In contrast, in our paper, the reconstruction objective for HeartLang's VQ-HBR process is the original heartbeat. The entire goal of the VQ-HBR process is to construct an ECG vocabulary where morphologically similar and contextually consistent heartbeats are mapped to the same collective ECG word. Fundamentally, LaBraM remains a traditional time-frequency modeling approach, while our method focuses on a morphology-semantic expression level of modeling.
> - **Effectiveness of QRS-based Temporal Embedding:** In our analysis of the HeartLang architecture, ablation experiments reveal the simplicity and effectiveness of the QRS-based Temporal Embedding. The indices for HeartLang's Temporal Embedding primarily originate from a byproduct of the QRSTokenizer—the QRS complex index. In Section 5.4 of the ablation studies, we find that the framework with only Temporal Embedding consistently achieves the second-highest or even the highest performance. This simple and effective model design offers inspiration for future research in the ECG domain, particularly by providing a feasible solution for injecting temporal information into subsequent ELP studies.
> - **Differences in Model Architecture Details:** The backbone network of HeartLang does not employ a Temporal Encoder to capture dynamic information. Instead, we use a simple and efficient QRS-based Temporal Embedding, whose effectiveness has been demonstrated in ablation experiments. Additionally, HeartLang does not utilize Symmetric Masking during the pretraining phase to improve computational efficiency. We aim for the performance improvement of our method to stem from the proposed ECG language modeling perspective rather than auxiliary techniques.
>
> We acknowledge being inspired by the LaBraM paper you mentioned and have appropriately cited it in the main text. As you pointed out, our work is more focused on innovating the concept of data slicing for ECG signals. We treat ECG signals as a language and model them using approaches from natural language processing. Given that most tasks in the ECG research are classification tasks, we adopt a BERT-like architecture. There have already been some successful efforts to apply the BERT architecture to other domains, such as BEiT v2 [3] (computer vision) and LaBraM [4] (EEG signals). Our aim is to apply the BERT architecture to the ECG research field, which has resulted in a similar model architecture. However, there will still be differences, such as the aforementioned points.
>
> Thank you again for your valuable feedback and insights.
>
>
>
> [1] Kawin Ethayarajh. 2019. How Contextual are Contextualized Word Representations? Comparing the Geometry of BERT, ELMo, and GPT-2 Embeddings. EMNLP 2019.
>
> [2] Oh, Jungwoo, et al. "Lead-agnostic self-supervised learning for local and global representations of electrocardiogram." Conference on Health, Inference, and Learning. PMLR, 2022.
>
> [3] Peng, Zhiliang, et al. "Beit v2: Masked image modeling with vector-quantized visual tokenizers." arXiv preprint arXiv:2208.06366 (2022).
>
> [4] Jiang, Wei-Bang, Li-Ming Zhao, and Bao-Liang Lu. "Large brain model for learning generic representations with tremendous EEG data in BCI."ICLR 2024.

---

> > ### Comment · Reviewer_xUd7 · 2024-11-27
> >
> > The authors provided explanations to the questions and updated the manuscript accordingly which looks better now.

---

> > > ### Author Response · Authors · 2024-11-27
> > > **Sincere Thanks for Reviewer xUd7**
> > >
> > > We are deeply grateful for your careful review and valuable feedback. Your expertise and detailed evaluation have contributed significantly to the improvement of our manuscript.

---

### Official Review · Reviewer_HUR2 · 2024-10-30

**Soundness:** 3
**Presentation:** 3
**Contribution:** 2
**Rating:** 8
**Confidence:** 3

**Summary:**

This work addresses limitations in deep learning for ECG analysis (usually based on fixed-size and fixed-step time windows) by proposing a novel approach that treats ECG signals like language, where heartbeats are "words" and rhythms are "sentences". The authors introduce the QRS-Tokenizer to segment ECG signals into meaningful "sentences" and present HeartLang, a self-supervised framework that learns ECG representations at the form and rhythm levels. A comprehensive heartbeat-based ECG vocabulary is provided. This approach shows strong performance across 6 ECG data sets.

**Strengths:**

- Original idea to treat ECG signal as language in form of sentences and words
- Straightforward approach to tokenize the ECG signal into words and sentences (QRS-Tokenizer)
- Large ECG vocabulary - also useful and interpretable from a clinical perspective
- Systematic comparison of fixed-size and fixed-step time windows vs. language approach
- Benchmarking of results (using standard settings of MERL)
- Informative ablation study regarding spatial / temporal embeddings, pretraining and vocabulary set

**Weaknesses:**

- A similar version of this manuscript has already been accepted: https://openreview.net/pdf?id=sdczcOS3n9 (at KDD-AIDSH 2024)

**Questions:**

- To what extent does the current manuscript differ to the manuscript accepted at KDD-AIDSH 2024? What are the novel findings?

-> Has been clarified in the revisions, thank you!

---

> ### Author Response · Authors · 2024-11-19
> **Response to Reviewer HUR2 (1/2)**
>
> We are extremely grateful for your review of the manuscript. To begin, the manuscript you referenced is a workshop paper. These two manuscripts differ significantly in their experimental setups, content, and findings.
>
> > **Q1: What extent does the current manuscript differ to the manuscript accepted at KDD-AIDSH 2024?**
>
> **A1:** The current manuscript differs significantly from the manuscript you mentioned in terms of experimental setup, content, and findings.
>
> - In terms of experimental setup, the current manuscript uses the MIMIC-IV-ECG dataset for pretraining and the PTBXL-Superclass, PTBXL-Subclass, PTBXL-Form, PTBXL-Rhythm, CPSC2018, and Chapman-Shaoxing-Ningbo (CSN) datasets for downstream task validation. In contrast, the manuscript you mentioned only used the PTB-XL series datasets for both pretraining and downstream validation. The MIMIC-IV-ECG dataset comprises 800,035 12-lead ECGs collected from 161,352 subjects, making it one of the largest publicly available ECG datasets to date. Meanwhile, the PTB-XL dataset includes only 21,837 12-lead ECGs collected from 18,885 subjects. For downstream task validation, we utilized six datasets—PTBXL-Superclass, PTBXL-Subclass, PTBXL-Form, PTBXL-Rhythm, CPSC2018, and CSN—covering over 100 cardiac conditions. The current manuscript follows the recommendations from MERL (ICML 2024) regarding upstream and downstream datasets to ensure fair comparisons with 10 other self-supervised learning methods.
> - In terms of experimental content, the current manuscript differs by including validation on additional downstream datasets such as the CPSC2018 and CSN datasets (Section 5.1), an evaluation of the proposed perspective on signal segmentation (Section 5.2), a detailed discussion on ECG Vocabulary (Section 5.3), and comprehensive ablation studies (Section 5.4). In Section 5.2, we validate the advantages of the "Heartbeat as Words and Rhythms as Sentences" perspective. Compared to traditional fixed-time window segmentation methods, our approach demonstrates an average improvement of 5.36 in macro AUC across downstream tasks. Notably, in disease superclass and subclass datasets, this perspective increases the macro AUC by an average of 8.38. In Section 5.3, the current manuscript provides a detailed discussion of the ECG Vocabulary, revealing that even similar heartbeat morphologies can yield different semantic representations based on context. This finding marks a significant advancement over previous ECG Language Processing (ELP) studies, where vocabulary construction typically relied on pre-clustering, often lacking contextual information and resulting in a limited vocabulary with less rich semantic representation. In Section 5.4, a comprehensive ablation study on the HeartLang structure demonstrates the importance of each component within HeartLang.
> - The current manuscript thus differs significantly from the manuscript you mentioned in all these areas, which you can refer to for review based on the content outlined above.

---

> ### Author Response · Authors · 2024-11-19
> **Response to Reviewer HUR2 (2/2)**
>
> > **Q2: What are the novel findings?**
>
> **A2:** Our novel findings primarily focus on the analysis of the ECG vocabulary and the HeartLang architecture.
>
> - For the ECG vocabulary, a novel finding in the current manuscript is that in our constructed vocabulary, even similar heartbeat morphologies can yield different semantic representations based on contextual information. In natural language processing, context plays a crucial role in shaping the semantic representation of specific words. For example, the word "run" can be either a noun or a verb, with its meaning dependent on surrounding context. In our work, the vector-quantized heartbeat reconstruction training dynamically constructs the vocabulary, with spatio-temporal embedding and position embedding allowing our vocabulary to incorporate contextual information. In contrast, previous ECG Language Processing (ELP) research has commonly used K-means clustering for vocabulary construction, which captures only heartbeat morphology without contextual information. This discovery significantly enriches the semantic depth of the vocabulary in ELP, contributing to advancements in the field.
> - For the HeartLang architecture, we find that the Temporal Embedding based on QRS waves is both simple and effective in our ablation studies. The Temporal Embedding in HeartLang is indexed primarily from the by-product of the QRS-Tokenizer—the QRS complex index. Specifically, we divide each 10-second ECG signal into 10 intervals, and for each individual ECG word, we assign the temporal embedding corresponding to the interval where its QRS complex index is located. In the ablation study in Section 5.4, we observe that when the framework includes only Temporal Embedding, it often achieves the second-highest or even the highest performance. This simple and effective model architecture could inspire future research in the ECG field, especially by offering a feasible solution for incorporating temporal information in subsequent ELP studies.
>
> Thank you again for your valuable feedback and insights.

---

> > ### Comment · Reviewer_HUR2 · 2024-11-25
> >
> > Thanks for clarifying my point regarding the previous publication and the novelty! I already adapted my score.

---

> > > ### Author Response · Authors · 2024-11-27
> > > **Sincere Thanks for Reviewer HUR2**
> > >
> > > Thank you very much for taking the time to review our manuscript. We greatly appreciate your thorough evaluation and the acknowledgment of our work. Your positive feedback and support serve as an encouragement to our research efforts.

---

### Official Review · Reviewer_6RzM · 2024-11-03

**Soundness:** 3
**Presentation:** 3
**Contribution:** 3
**Rating:** 8
**Confidence:** 5

**Summary:**

This paper introduces HeartLang, a self-supervised learning framework designed for analyzing electrocardiogram (ECG) signals by treating them like a language, with heartbeats as "words" and rhythms as "sentences." This approach diverges from conventional ECG analysis by using a language model perspective that preserves the form and rhythm characteristics of ECG signals, enhancing representation learning. Key contributions include:

1. QRS-Tokenizer: This tool segments ECG signals into meaningful "sentences" by identifying individual heartbeats (treated as words).
2. ST-ECGFormer: A spatio-temporal network designed to capture and process the temporal and spatial features of ECG data.
3. ECG Vocabulary: The largest heartbeat-based vocabulary to date, with 5,394 distinct heartbeat types for diverse cardiac conditions.
4. Masked ECG Sentence Pre-Training: A pre-training method that learns rhythm-level representations by masking parts of ECG "sentences."

HeartLang was tested on three public datasets, showing improved performance over other ECG self-supervised learning methods in areas such as general representation learning and downstream tasks, such as classification. This approach opens up a language-based approach for ECG research.

**Strengths:**

This study on the HeartLang self-supervised learning framework has several strengths that make it a valuable contribution to the field of ECG analysis:
- Innovative ECG Language Perspective: The approach of treating ECG signals as a language, where heartbeats are "words" and rhythms are "sentences," is novel. This shift from conventional time-series analysis captures the unique form and rhythm characteristics of ECG signals, which can be critical for diagnosing cardiac conditions.
- Introduction of the QRS-Tokenizer: This component is specially designed to segment ECG signals into semantically meaningful units, allowing the model to focus on clinically relevant patterns in the data. The QRS-Tokenizer aligns the structure of the data with natural language processing (NLP) techniques, enhancing the depth and utility of the learned representations.
- Transformer-Based Architecture with ST-ECGFormer: The framework’s backbone, ST-ECGFormer, is a transformer-based model specifically designed to capture spatio-temporal features in ECG data. This model helps HeartLang learn complex temporal and spatial dependencies in the signals, enhancing the quality of the extracted features for further tasks.
- Flexibility in Downstream Applications: HeartLang’s pre-trained representations can be adapted for various downstream tasks, such as classification, providing a flexible tool for researchers and clinicians to build specific diagnostic models based on general ECG representations.

These strengths make HeartLang a promising advancement for ECG data analysis, particularly in enhancing the interpretability and applicability of ECG-based machine learning models in clinical diagnostics.

**Weaknesses:**

This approach would be effective in cases where the QRS waveform is recorded on an electrocardiogram, i.e. in cases of ST abnormalities due to ischemic heart disease or bundle branch block. However, many heart diseases cause significant irregularities in the heart rhythm itself, and these conditions are generally more directly linked to life. It is difficult to think of the HeartLang approach as being suitable for these conditions, and it is thought that it will be necessary to supplement it with other approaches in order to cover the whole of electrocardiography or heart disease.
The scope of this study may be narrow than the general interest of ICLR main conference.

**Questions:**

The approach of this research, which treats each heartbeat as a word and the rhythm of the time series as a sentence, is a very good idea for bringing the success of large-scale language models in natural language to electrocardiograms, and as described in this paper, it has also been successfully implemented. Continuing from the above weakness section, it would be even better if there was a clear discussion of the limitations in the main text.

---

> ### Author Response · Authors · 2024-11-19
> **Response to Reviewer 6RzM**
>
> We truly appreciate your encouragement, careful review, and valuable suggestion. You have raised an important issue. We agree with your comments and have modified our manuscript accordingly.
>
> > **Q1: The approach of this research, which treats each heartbeat as a word and the rhythm of the time series as a sentence, is a very good idea for bringing the success of large-scale language models in natural language to electrocardiograms, and as described in this paper, it has also been successfully implemented. Continuing from the above weakness section, it would be even better if there was a clear discussion of the limitations in the main text.**
>
> **A1:** Your concerns are valid, and HeartLang maybe exhibit some performance decline in classifying certain heart conditions with less pronounced QRS waves. We will clarify this limitation in our discussion. However, as our current manuscript has already reached the 10-page limit imposed by ICLR for the main text, we have placed this discussion in the appendix to facilitate reference for future research.
>
> Thank you again for your valuable feedback and insights.

---

> > ### Comment · Reviewer_6RzM · 2024-11-27
> >
> > The concerns were shared and the authors took actions in the appendix section, thus I maintain my score.

---

> > > ### Author Response · Authors · 2024-11-27
> > > **Sincere Thanks for Reviewer 6RzM**
> > >
> > > We deeply appreciate your insightful and constructive feedback. Your comments have guided us to refine our manuscript and ensure greater clarity and rigor in our presentation.

---

### Official Review · Reviewer_5txW · 2024-11-04

**Soundness:** 3
**Presentation:** 3
**Contribution:** 3
**Rating:** 6
**Confidence:** 4

**Summary:**

The paper introduces HeartLang, a self-supervised learning framework for ECG language processing that conceptualizes heartbeats as "words" and rhythms as "sentences" to better capture ECG signal semantics. The framework includes the QRS-Tokenizer, which segments ECG signals into meaningful units, and, a transformer-based model for learning spatiotemporal and semantic representations. The author's proposed approach, which uses heartbeat reconstruction and masked sentence pre-training and conducts experiments on six ECG datasets, outperforming other models in tasks such as rhythm and form classification. This work positions ECG data processing within a language-like framework, aiming to enrich model interpretability and generalizability for clinical applications.

**Strengths:**

The paper is well-written with good clarity and coherence making the paper easy to read and comprehend. Most work on ECG either focus on beats or on rhythms making it challenging to learn both intra-beat and inter-beat features together which is where this contribution shines. The contribution is significant and interest with the concept of beats as words and rhythms as sentences which would allow learning more comprehensive latent features about the beats itself as well as the relation between the beats similar to human language. The masked training approach although not novel is a novel application with the concept of beats as words and rhythms as sentences and achieved significantly improved performance on popular datasets.

**Weaknesses:**

It is mentioned that the QRS-Tokenizer is used to segment the raw ECG signals into ECG sentences but as per my understanding the QRS tokenizer should be used to segment into each ECG beats aka words which is confusing. The application of HeartLang at this point is limited and would require investigation on downstream tasks along with comparative studies.

**Questions:**

It is mentioned that the QRS-Tokenizer is used to segment the raw ECG signals into ECG sentences but as per my understanding the QRS tokenizer should be used to segment into each ECG beats aka words?

---

> ### Author Response · Authors · 2024-11-19
> **Response to Reviewer 5txW**
>
> We are extremely grateful for your review of the manuscript. You have raised an important issue. We agree with your comments and have modified our manuscript accordingly.
>
> > **Q1: It is mentioned that the QRS-Tokenizer is used to segment the raw ECG signals into ECG sentences but as per my understanding the QRS tokenizer should be used to segment into each ECG beats aka words?**
>
> **A1:** Yes, your understanding is correct. Thank you for your careful observation and we apologize for any confusion caused by our wording.
>
> Indeed, the process described in Section 3.1, “Generating ECG Sentences Using the QRS-Tokenizer,” consists of two main steps: QRS Detection and Generating ECG Sentences. During the QRS Detection step, the raw ECG signal undergoes QRS wave detection to determine the position of each heartbeat and segment individual ECG words (heartbeats). In the subsequent Generating ECG Sentences step, we concatenate these words to form an ECG sentence by rules.
>
> Following your suggestion, we will revise the manuscript to adjust statements like “QRS-Tokenizer is used to segment the raw ECG signals into ECG sentences” to “QRS-Tokenizer is used to generate the ECG sentences from the raw ECG signals.”
>
> Additionally, regarding the potential limitation you mentioned about the broader application of HeartLang, please refer to Section 5.1, "Evaluation on Linear Probing," and Table 1. Our study follows the downstream task validation approach recommended by MERL (ICML 2024), utilizing six downstream datasets—PTBXL-Superclass, PTBXL-Subclass, PTBXL-Form, PTBXL-Rhythm, CPSC2018, and CSN—to cover over 100 distinct cardiac conditions. HeartLang has shown strong competitiveness compared to 10 other self-supervised methods. The extensive downstream validation and strong competitiveness demonstrate relatively broad applicability of HeartLang.
>
> Thank you again for your valuable feedback and insights.

---

> ### Author Response · Authors · 2024-11-27
> **Sincere Thanks for Reviewer 5txW**
>
> We are truly grateful for your professional and meticulous review. Your observations and suggestions have significantly contributed to the improvement of our manuscript.

---

### Meta-Review · Area_Chair_gZei · 2024-12-20

**Metareview:**

This paper has a simple three-step architecture to model ECG signals. First, the creation of a QRSTokenizer that uses basic signal processing to identify QRS signals within the ECG data, then an attention based transformer that combines spatial and temporal information. The output of the transformer is then reconstructed using a learned ECG codebook which is then reconstructed into a signal via an ECG codebook. The entire process is trained end to end using a mean squared error loss and a combination of losses to regularize learning the codebook. Overall, the reviewers found the work simple, well motivated and empirically performant. I think the use of the ECG codebook is clever and could be useful in a variety of downstream tasks well beyond its use in the tasks herein. Their finding that "even similar heartbeat morphologies can yield different semantic representations based on contextual information" I believe will be of independent interest to cardiologists studying diseases where such phenomenon are observed clinically.

**Additional Comments On Reviewer Discussion:**

During the discussion period, there were three ablation studies that were discussed that were added into the supplementary results:
a] understanding the impact of the vocabulary size on performance in downstream tasks via ablation experiments
b] understanding the impact of the number of leads on downstream performance via ablation experiments
c] clarification of novelty of the manuscript.

---

### Decision · Program_Chairs · 2025-01-22

Accept (Poster)